# Neural Message Passing for Multi-Label Classification

## Abstract

Multi-label classification (MLC) is the task of assigning a set of target labels for a given sample. Modeling the combinatorial label interactions in MLC has been a long-haul challenge. Recurrent neural network (RNN) based encoder-decoder models have shown state-of-the-art performance for solving MLC. However, the sequential nature of modeling label dependencies through an RNN limits its ability in parallel computation, predicting dense labels, and providing interpretable results. In this paper, we propose Message Passing Encoder-Decoder (MPED) Networks, aiming to provide fast, accurate, and interpretable MLC. MPED networks model the joint prediction of labels by replacing all RNNs in the encoder-decoder architecture with message passing mechanisms and dispense with autoregressive inference entirely. The proposed models are simple, fast, accurate, interpretable, and structure-agnostic (can be used on known or unknown structured data). Experiments on seven real-world MLC datasets show the proposed models outperform autoregressive RNN models across five different metrics with a significant speedup during training and testing time.

## 1 Introduction

Multi-label classification (MLC) is receiving increasing attention in tasks such as text categorization and image classification. Accurate and scalable MLC methods are in urgent need for applications like assigning topics to web articles, classifying objects in an image, or identifying binding proteins on DNA. The most common and straightforward MLC method is the binary relevance (BR) approach that considers multiple target labels independently (Tsoumakas and Katakis, 2006). However, in many MLC tasks there is a clear dependency structure among labels, which BR methods ignore.

Accordingly, probabilistic classifier chain (PCC) models were proposed to model label dependencies and formulate MLC in an autoregressive sequential prediction manner (Read et al., 2009). One notable work in the PCC category was from Nam et al. (2017) which implemented a classifier chain using a recurrent neural network (RNN) based sequence to sequence (Seq2Seq) architecture, Seq2Seq MLC. This model uses an encoder RNN encoding elements of an input sequence, a decoder RNN predicting output labels one after another, and beam search that computes the probability of the next $T$ predictions of labels and then chooses the proposal with the max combined probability.

However, the main drawback of classifier chain models is that their inherently sequential nature precludes parallelization during training and inference. This can be detrimental when there are a large number of positive labels as the classifier chain has to sequentially predict each label, and often requires beam search to obtain the optimal set. Aside from time-cost disadvantages, PCC methods have several other drawbacks. First, PCC methods require a defined ordering of labels for the sequential prediction, but MLC output labels are an unordered set, and the chosen order can lead to prediction instability (Nam et al., 2017). Secondly, even if the optimal ordering is known, PCC methods struggle to accurately capture long-range dependencies among labels in cases where the number of positive labels is large (i.e., dense labels). For example, the Delicious dataset has a median of 19 positive labels per sample, so it can be difficult to correctly predict the labels at the end of the prediction chain. Lastly, many real-world applications prefer interpretable predictors. For instance, in the task of predicting which proteins (labels) will bind to a DNA sequence based binding site, users care about how a prediction is made and how the interactions among labels influence the predictions[1].

---

[1] An important task is modelling what is known as "co-binding" effects, where one protein will *only* bind if there is another specific protein already binding, or similarly will not bind if there is another already binding.

Message Passing Neural Networks (MPNNs) (Gilmer et al., 2017) introduce a class of methods that model joint dependencies of variables using neural message passing rather than an explicit representation such as a probabilistic classifier chain. Message passing allows for efficient inference by modelling conditional independence where the same local update procedure is applied iteratively to propagate information across variables. MPNNs provide a flexible method for modeling multiple variables jointly which have no explicit ordering (and can be modified to incorporate an order, as explained in section 3). To handle the drawbacks of BR and PCC methods, we propose a modified version of MPNNs for MLC by modeling interactions between labels using neural message passing.

We introduce Message Passing Encoder-Decoder (MPED) Networks aiming to provide fast, accurate, and interpretable multi-label predictions. The key idea is to replace RNNs and to rely on neural message passing entirely to draw global dependencies between input components, between labels and input components, and between labels. The proposed MPED networks allow for significantly more parallelization in training and testing. The main contributions of this paper are:

- **Novel approach for MLC.** To the authors' best knowledge, MPED is the first work using neural message passing for MLC.
- **Accurate MLC**. Our model achieves similar, or better performance compared to the previous state of the art across five different MLC metrics. We validate our model on seven MLC datasets which cover a wide spectrum of input data structure: sequences (English text, DNA), tabular (bag-of-words), and graph (drug molecules), as well as output label structure: unknown and graph.
- **Fast**. Empirically our model achieves an average 1.7x speedup over the autoregressive seq2seq MLC at training time and an average 5x speedup over its testing time.
- **Interpretable**. Although deep-learning based systems have widely been viewed as "black boxes" due to their complexity, our attention based MPED models provide a straightforward way to explain label to label, input to label, and feature to feature dependencies.

## 2 METHOD: MPED NETWORKS

### 2.1 BACKGROUND: MESSAGE PASSING NEURAL NETWORKS

Message Passing Neural Networks (MPNNs) (Gilmer et al., 2017) are a generalization of graph neural networks (GNNs) (Gori et al., 2005; Scarselli et al., 2009), where variables are represented as nodes on a graph $G$ and joint dependencies are modelled using message passing rather than explicit representations, which allows for efficient inference. MPNNs model the joint dependencies using message function $M^t$ and node update function $U^t$ for $T$ time steps, where $t$ is the current time step. The hidden state $\boldsymbol{v}_i^t \in \mathbb{R}^d$ of node $i \in G$ is updated based on messages $\boldsymbol{m}_i^t$ from its neighboring nodes $\{\boldsymbol{v}_{j \in \mathcal{N}(i)}^t\}$ defined by neighborhood $\mathcal{N}(i)$:

$$\boldsymbol{m}_i^t = \sum_{j \in \mathcal{N}(i)} M^t(\boldsymbol{v}_i^t, \boldsymbol{v}_j^t), \tag{1}$$

$$\boldsymbol{v}_i^{t+1} = U^t(\boldsymbol{v}_i^t, \boldsymbol{m}_i^t). \tag{2}$$

After $T$ updates, a readout function $R$ is used on the updated nodes for a prediction (e.g., node classification or graph classification) on the graph $G$.

Many possibilities exist for functions $M^t$ and $U^t$. For example, one can pass messages using neural attention in which nodes are able to attend over their neighborhoods differentially (Veličković et al., 2017). This allows for the network to learn different weights for different nodes in a neighborhood, without depending on knowing the graph structure a priori. In this formulation, messages for node $\boldsymbol{v}_i^t$ are obtained by a weighted sum of all neighboring nodes $\{\boldsymbol{v}_{j \in \mathcal{N}(i)}^t\}$ where the weights are obtained by attention (Bahdanau et al., 2014). In our implementation, we implement neural message passing with attention. In the rest of the paper, we use "graph attention" and "neural message passing" interchangeably. Neural message passing with attention works as follows.

Attention weights $\alpha_{ij}^t$ for pair of nodes $(\boldsymbol{v}_i^t, \boldsymbol{v}_j^t)$ are computed using some attention function $a(\cdot)$:

$$e_{ij}^t = a(\boldsymbol{v}_i^t, \boldsymbol{v}_j^t), \tag{3}$$

where $e_{ij}^t$ represents the importance of node $j$ for node $i$. For our attention function, we used a scaled dot product with node-wise linear transformations $\mathbf{W}^q \in \mathbb{R}^{d \times d}$ on node $\boldsymbol{v}_i^t$ and $\mathbf{W}^u \in \mathbb{R}^{d \times d}$ on

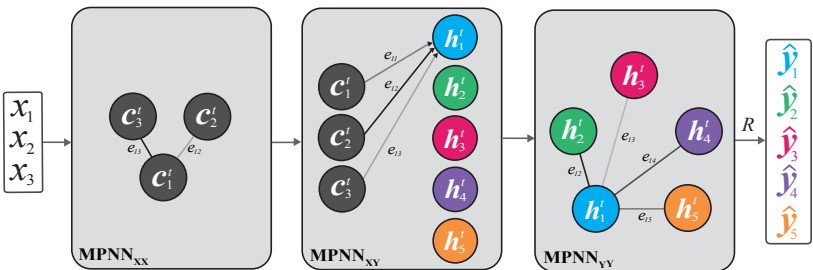

Figure 1: **MPED Networks.** Given input **x**, we encode its components $\{x_1, x_2, x_3\}$ as embedded input nodes $\{c_1^t, c_2^t, c_3^t\}$ of encoder graph $G_{ENC}$. Similarly, we encode labels $\{y_1, y_2, ..., y_5\}$ as embedded label nodes $\{h_1^t, h_2^t, ..., h_5^t\}$ of decoder graph $G_{DEC}$. MPNN$_{\text{xx}}$ is used to pass messages between the input nodes and update input nodes. MPNN$_{\text{xy}}$ is used to pass messages from the input nodes to the labels nodes and update the label nodes. MPNN$_{\text{yy}}$ is used to pass messages between the label nodes and update label nodes. Finally, readout function $R$ performs node-level classification on label nodes to make binary label predictions $\{\hat{y}_1, \hat{y}_2, ..., \hat{y}_5\}$.

node $v_j^t$. Scaling by $\sqrt{d}$ is used to mitigate training issues (Vaswani et al., 2017):

$$a(v_i^t, v_j^t) = \frac{(\mathbf{W}^q v_i^t)^\top (\mathbf{W}^u v_j^t)}{\sqrt{d}}. \tag{4}$$

Attention coefficients $e_{ij}^t$ are then normalized across all neighboring nodes of node $i$ using a softmax function:

$$\alpha_{ij}^t = \text{softmax}_j(e_{ij}^t) = \alpha_{ij}^t = \frac{\exp(e_{ij}^t)}{\sum_{k \in \mathcal{N}(i)} \exp(e_{ik}^t)}. \tag{5}$$

In our method, we use a so called attention message function $M_{\text{atn}}^t$ to produce the message from node $j$ to node $i$ using the learned attention weights $\alpha_{ij}^t$ and another transformation matrix $\mathbf{W}^v \in \mathbb{R}^{d \times d}$. Then we compute the full message $m_i^t$ by linearly combining messages from all nodes $j \in \mathcal{N}(i)$ with a residual connection on the current $v_i^t$:

$$M_{\text{atn}}(v_i^t, v_j^t; \boldsymbol{W}) = \alpha_{ij}^t \mathbf{W}^v v_j^t, \tag{6}$$

$$m_i^t = v_i^t + \sum_{j \in \mathcal{N}(i)} M_{\text{atn}}(v_i^t, v_j^t; \boldsymbol{W}). \tag{7}$$

Lastly, node $v_i^t$ is updated to state $v_i^{t+1}$ using message $m_i^t$ by a multi-layer perceptron (MLP) update function $U_{\text{mlp}}$ with matrices $\mathbf{W}^r \in \mathbb{R}^{2d \times d}$, $\mathbf{W}^b \in \mathbb{R}^{2d \times d}$, and a residual connection on $m_i^t$:

$$U_{\text{mlp}}(m_i^t; \boldsymbol{W}) = \text{ReLU}(\mathbf{W}^r m_i^t + b_1)^\top \mathbf{W}^b + b_2, \tag{8}$$

$$v_i^{t+1} = m_i^t + U_{\text{mlp}}(m_i^t; \boldsymbol{W}). \tag{9}$$

It is important to note that matrices $\mathbf{W}$ are shared (i.e., separately applied) across all nodes. This can be viewed as 1-dimensional convolution with kernel and stride sizes of 1. Weight sharing across nodes is a key aspect of MPNNs, where node dependencies are learned in an order-invariant manner.

## 2.2 MPED: Message Passing Encoder-Decoder Network for MLC

**Notations:** We define the following notations, used throughout the paper. Let $\mathcal{D} = \{(\mathbf{x}_n, \mathbf{y}_n)\}_{n=1}^N$ be the set of data samples with inputs $\mathbf{x} \in X$ and outputs $\mathbf{y} \in Y$. Inputs $\mathbf{x}$ are a (possibly ordered) set of $S$ components $\{x_1, x_2, ..., x_S\}$, and outputs $\mathbf{y}$ are a set of $L$ labels $\{y_1, y_2, ..., y_L\}$. MLC involves predicting the set of binary labels $\{y_1, y_2, ..., y_L\}, y_i \in \{0, 1\}$ given input $\mathbf{x}$.

Input features are represented as embedded vectors $\{c_1^{t=0}, c_2^{t=0}, ..., c_S^{t=0}\}, c_i^t \in \mathbb{R}^d$, using embedding matrix $\mathbf{W}^x \in \mathbb{R}^{\delta \times d}$, where $d$ is the embedding size, $\delta$ is the vocabulary size, and $t$ represents the 'state' of the embedding after $t$ updates. Similarly, labels are represented as an embedded vectors $\{h_1^{t=0}, h_2^{t=0}, ..., h_L^{t=0}\}, h_i^t \in \mathbb{R}^d$, using embedding matrix $\mathbf{W}^y \in \mathbb{R}^{L \times d}$, where $L$ is the number of labels, and $t$ represents the 'state' of the embedding after $t$ updates.

**Setup:** In MLC, each output label is determined by a joint probability of other labels and the input features. Our goal is to achieve the performance of explicit joint probability methods such as PCCs (Eqs. 22 and 23), at the test speed of BR methods (Eq. 21).

We introduce Message Passing Encoder-Decoder (MPED) networks, where we formulate MLC using an encoder-decoder architecture. In MPED Networks, input components are represented as nodes in encoder graph $G_{ENC}$ using embedding vectors $\{c_{1:S}^t\}$, and labels are represented as nodes in decoder graph $G_{DEC}$ using embedding vectors $\{h_{1:L}^t\}$. MPED networks use three MPNN modules with attention to pass messages within $G_{ENC}$, from $G_{ENC}$ to $G_{DEC}$, and within $G_{DEC}$ to model the joint prediction of labels.

The first module, MPNN$_{xx}$, is used to update input component nodes $\{c_{1:S}^t\}$ by passing messages within the encoder (between input nodes). The second module, MPNN$_{xy}$, is used to update output label nodes $\{h_{1:L}^t\}$ by passing messages from the encoder to decoder (from input nodes $\{c_{1:S}^t\}$ to output nodes $\{h_{1:L}^t\}$). The third module, MPNN$_{yy}$, is used to update output label nodes $\{h_{1:L}^t\}$ by passing messages within the decoder (between label nodes). Once messages have been passed to update input and label nodes, a readout function $R$ is then used on the label nodes to make a binary classification prediction for each label, $\{\hat{y}_1, \hat{y}_2, ..., \hat{y}_L\}$. An overview of our model is shown in Fig. 1.

### 2.3 ENCODER: MODELING FEATURE INTERACTIONS VIA INPUT COMPONENT TO COMPONENT MESSAGE PASSING

For a particular input $\mathbf{x}$, we first assume that the input features $\{x_{1:S}\}$ are nodes on a graph, we call $G_{ENC}$. $G_{ENC} = (V, E)$, $V = \{x_{1:S}\}$, and $E$ includes all undirected pairwise edges connecting node $c_i$ and node $c_j$. MPNN$_{xx}$, parameterized by $\mathbf{W}_{xx}$, is used to pass messages between the input embeddings in order to update their states. $x_i$ can be any component of a particular input (e.g. words in a sentence, patches of an image, nodes of a known graph, or tabular features).

Nodes on $G_{ENC}$ are represented as embedding vectors $\{c_1^t, c_2^t, ..., c_S^t\}$, where the initial states $\{c_{1:S}^0\}$ are obtained using embedding matrix $\mathbf{W}^x$ on the input components $\{x_1, x_2, ..., x_S\}$. The embeddings are then updated by MPNN$_{xx}$ using message passing for $T$ time steps to produce $\{c_1^T, c_2^T, ..., c_S^T\}$.

To update input embedding $c_i^t$, MPNN$_{xx}$ uses attention message function $M_{\text{atn}}^t$ (Eq. 6) on all neighboring input embeddings $\{c_{j \in \mathcal{N}(i)}^t\}$ to produce messages $m_i^t$, and MLP update function $U_{\text{mlp}}$ (Eq. 8) to produce updated embedding $c_i^{t+1}$:

$$m_i^t = c_i^t + \sum_{j \in \mathcal{N}(i)} M_{\text{atn}}(c_i^t, c_j^t; \mathbf{W}_{xx}), \tag{10}$$

$$c_i^{t+1} = m_i^t + U_{\text{mlp}}(m_i^t; \mathbf{W}_{xx}). \tag{11}$$

If there exists a known $G_{ENC}$ graph, message $m_i^t$ for node $i$ is computed using its neighboring nodes $j \in \mathcal{N}(i)$, where the neighbors $\mathcal{N}(i)$ are defined by the graph. If there is no known graph, we assume a fully connected $G_{ENC}$ graph, which means $\mathcal{N}(i) = \{j \neq i\}$. Inputs with a sequential ordering can be modelled as a fully connected graph using positional embeddings (Battaglia et al., 2018).

### 2.4 DECODER

Similar to the input components in the encoder, we assume that the labels $\{y_{1:L}\}$ are nodes on a decoder graph called $G_{DEC}$. Nodes on $G_{DEC}$ are represented as embedding vectors $\{h_1^t, h_2^t, ..., h_L^t\}$, where the initial states $\{h_{1:L}^0\}$ are obtained using label embedding matrix $\mathbf{W}^y$. The decoder MPNNs update the label embeddings $\{h_{1:L}^t\}$ by passing messages from the encoder to the decoder, and then pass messages within the decoder. MPNN$_{xy}$, is used to pass messages from input embeddings $\{c_{1:S}^T\}$ to label embeddings, and then MPNN$_{yy}$ is used to pass messages between label embeddings.

#### 2.4.1 UPDATING LABEL EMBEDDINGS THROUGH INPUT TO LABEL MESSAGE PASSING

In order to update the label nodes given a particular input $\mathbf{x}$, the decoder uses MPNN$_{xy}$, parameterized by $\mathbf{W}_{xy}$, to pass messages from input $\mathbf{x}$ to labels $\mathbf{y}$. At the equation level, this module is identical to MPNN$_{xx}$ except that it updates the $i^{th}$ label node's embedding $h_i$ using the embeddings of all the components of an input. That is, we update each $h_i^t$ by using a weighted sum of all input embeddings $\{c_{1:S}^T\}$, in which the weights represent how important an input component is to the $i^{th}$ label node

and the weights are learned via attention. Messages are only passed from the encoder nodes to the decoder nodes, and not vice versa (i.e. encoder to decoder message passing is directed).

More specifically, to update label embedding $\boldsymbol{h}_i^t$, MPNN$_{xy}$ uses attention message function $M_{atn}^t$ on all embeddings of the input $\{\boldsymbol{c}_{1:S}^T\}$ to produce messages $\boldsymbol{m}_i^t$, and another MLP update function $U_{mlp}$ to produce the updated intermediate embedding state $\boldsymbol{h}_i^{t'}$:

$$\boldsymbol{m}_i^t = \boldsymbol{h}_i^t + \sum_{j=1}^{S} M_{atn}(\boldsymbol{h}_i^t, \boldsymbol{c}_j^T; \boldsymbol{W}_{xy}), \tag{12}$$

$$\boldsymbol{h}_i^{t'} = \boldsymbol{m}_i^t + U_{mlp}(\boldsymbol{m}_i^t; \boldsymbol{W}_{xy}). \tag{13}$$

The key advantage of input-to-label message passing with attention is that each label node can attend to different input nodes (e.g. different words in the sentence).

### 2.4.2 UPDATING LABEL EMBEDDINGS THROUGH LABEL TO LABEL MESSAGE PASSING

At this point, the decoder can make an independent prediction for each label conditioned on $\mathbf{x}$. However, in order to make more accurate predictions, we model interactions between the label nodes $\{\boldsymbol{h}_{1:L}^{t'}\}$ using message passing and update them accordingly. To do this we use a a third message passing module, MPNN$_{yy}$. At the equation level, this layer is identical to MPNN$_{xx}$ except that it replaces the input embeddings with label embeddings. In other words, label embedding $\boldsymbol{h}_i^{t'}$ is updated by a weighted combination through attention of all its neighbor label nodes $\{\boldsymbol{h}_{j \in \mathcal{N}(i)}^{t'}\}$.

To update each label embedding $\boldsymbol{h}_i^{t'}$, MPNN$_{yy}$ uses attention message function $M_{atn}^{t'}$ on all neighbor label embeddings $\{\boldsymbol{h}_{j \in \mathcal{N}(i)}^{t'}\}$ to produce message $\boldsymbol{m}_i^t$, and MLP update function $U_{mlp}^{t'}$ to compute updated embedding $\boldsymbol{h}_i^{t+1}$:

$$\boldsymbol{m}_i^{t'} = \boldsymbol{h}_i^{t'} + \sum_{j \in \mathcal{N}(i)} M_{atn}(\boldsymbol{h}_i^{t'}, \boldsymbol{h}_j^{t'}; \boldsymbol{W}_{yy}), \tag{14}$$

$$\boldsymbol{h}_i^{t+1} = \boldsymbol{m}_i^{t'} + U_{mlp}(\boldsymbol{h}_i^{t'}, \boldsymbol{m}_i^{t'}; \boldsymbol{W}_{yy}). \tag{15}$$

If there exists a known $G_{DEC}$ graph, message $\boldsymbol{m}_i^t$ for node $i$ is computed using its neighboring nodes $j \in \mathcal{N}(i)$, where the neighbors $\mathcal{N}(i)$ are defined by the graph. If there is no known $G_{DEC}$ graph, we assume a fully connected graph, which means $\mathcal{N}(i) = \{j \neq i\}$.

In our implementation, the label embeddings are updated by MPNN$_{xy}$ and MPNN$_{xx}$ for $T$ time steps to produce $\{\boldsymbol{h}_1^T, \boldsymbol{h}_2^T, ..., \boldsymbol{h}_L^T\}$.

### 2.5 READOUT LAYER (MLC PREDICTIONS FROM THE DECODER)

The last module of the decoder predicts each label $\{\hat{y}_1, ...\hat{y}_L\}$. A readout function $R$ projects each of the $L$ label embeddings $\boldsymbol{h}_i^T$ using projection matrix $\mathbf{W}^o \in \mathbb{R}^{L \times d}$, where row $\mathbf{W}_i^o \in \mathbb{R}^d$ is the learned output vector for label $i$. The calculated vector of size $L \times 1$ is then fed through an element-wise sigmoid function to produce probabilities of each label being positive:

$$\hat{y}_i = R(\boldsymbol{h}_i^T; \mathbf{W}^o) = \text{sigmoid}(\mathbf{W}_i^o \boldsymbol{h}_i^T). \tag{16}$$

In MPED networks we use binary the mean cross entropy on the individual label predictions to train the model. $p(y_i | \{y_{j \neq i}\}, \boldsymbol{c}_{1:S}^T; \mathbf{W})$ is approximated in MPED networks by jointly representing $\{y_{1:L}\}$ using message passing from $\{\boldsymbol{c}_{1:S}^T\}$ and from the embeddings of all neighboring labels $\{y_{j \in \mathcal{N}(i)}\}$.

### 2.6 MODEL DETAILS

**Multi-head Attention** In order to allow a particular node to attend to multiple other nodes (or multiple groups of nodes) at once, MPED uses multiple attention heads. Inspired by Vaswani et al. (2017), we use $K$ independent attention heads for each $\mathbf{W}^{\cdot}$ matrix during the message computation, where each matrix column $\mathbf{W}_j^{\cdot,k}$ is of dimension $d/K$. The generated representations are concatenated (denoted by $\|$) and linearly transformed by matrix $\mathbf{W}^z \in \mathbb{R}^{d \times d}$. Multi-head attention changes

message passing function $M_{\text{atn}}$, but update function $U_{\text{mlp}}$ stays the same.

$$e_{ij}^{t,k} = (\mathbf{W}^{q,k}\boldsymbol{v}_i^t)^\top(\mathbf{W}^{u,k}\boldsymbol{v}_j^t)/\sqrt{d} \tag{17}$$

$$\alpha_{ij}^{t,k} = \frac{\exp(e_{ij}^{t,k})}{\sum_{j\in\mathcal{N}(i)}\exp(e_{ij}^{t,k})} \tag{18}$$

$$M_{\text{atn}}^k(\boldsymbol{v}_i^t, \boldsymbol{v}_j^t; \boldsymbol{W}) = \alpha_{ij}^{t,k}\mathbf{W}^{v,k}\boldsymbol{v}_j^t, \tag{19}$$

$$\boldsymbol{m}_i^t = \boldsymbol{v}_i^t + \left(\left\|_{k=1}^K \left[\sum_{j\in\mathcal{N}(i)} M_{\text{atn}}^k(\boldsymbol{v}_i^t, \boldsymbol{v}_j^t; \boldsymbol{W})\right]\right)\mathbf{W}^z \tag{20}$$

**Graph Time Steps** To learn more complex relations among nodes, we compute $T$ time steps of embedding updates. This is essentially a stack of $T$ MPNN layers. Matrices $\mathbf{W}_\cdot^q, \mathbf{W}_\cdot^u, \mathbf{W}_\cdot^v, \mathbf{W}_\cdot^r, \mathbf{W}_\cdot^b, \mathbf{W}_\cdot^z$, are not shared across time steps (but are shared across nodes).

**Label Embedding Weight Sharing** To enforce each label's input embedding to correspond to that particular label, the label embedding matrix weights $\mathbf{W}^y$ are shared with the readout projection matrix $\mathbf{W}^o$. In other words, $\mathbf{W}^y$ is used to produce the initial node vectors for $G_{DEC}$, and then is used again to calculate the pre-sigmoid output values for each label, so $\mathbf{W}^o \equiv \mathbf{W}^y$. This was shown beneficial in Seq2Seq models for machine translation (Press and Wolf, 2016).

## 2.7 Advantages of MPED Models

**Speed.** In MPED models, the joint probability of labels isn't explicitly estimated using the chain rule. This enables making predictions in parallel and decreases test time drastically, especially when the number of labels is large. We model the joint probability implicitly using the MPED decoder, at the benefit of a substantial speedup. Time complexities of different types of models are compared in Table 1. The biggest advantage of MPED networks is constant training and testing times.

| Model | Training Time Cost (# sequential operations) | Testing Time Cost (# sequential operations) | Encoder Layer Space Cost | Decoder Layer Space Cost |
|---|---|---|---|---|
| RNN Seq2Seq | $O(S + \rho)$ | $O(\rho)$ | $O(S \cdot d^2)$ | $O(\rho \cdot d^2)$ |
| MPED | $O(1)$ | $O(1)$ | $O(S^2 \cdot d)$ | $O(L^2 \cdot d)$ |

Table 1: **Model Complexities.** $S$ represents the input sequence length, $\rho$ represents the number of positive labels, $L$ represents total number of labels, and $d$ is the size of the model's hidden state.

**Handling dense label predictions.** Motivated by the drawbacks of autoregressive models for MLC (Section 5.6), the proposed MPED model removes the dependencies on a chosen label ordering and beam search. This is particularly beneficial when the number of positive output labels is large (i.e. dense). MPED networks predict the output *set* of labels all at once, which is made possible by the fact that inference doesn't use a probabilistic chain, but there is still a representation of label dependencies via label to label attention. As an additional benefit, as noted by Belanger and McCallum (2016), it may be useful to maintain 'soft' predictions for each label in MLC. This is a major drawback of the PCC models which make 'hard' predictions of the positive labels, defaulting all other labels to 0.

**Flexibility** Many input or output types are instances where the relational structure is not made explicit, and must be inferred or assumed (e.g., text corpora, or MLC labels)(Battaglia et al., 2018). MPED networks allow for greater flexibility of input structures (known structure such as sequence or graph, or unknown such as tabular), or output structures (e.g., known graph vs unknown structure).

**Interpretability.** One advantage of MPED models is that interpretability is "built in" via neural attention. Specifically, we can visualize 3 aspects: input-to-input attention (input dependencies), input-to-label attention (input/label dependencies), and label-to-label attention (label dependencies).

## 2.8 Connecting to Related Topics

**Structured Output Predictions** The use of graph attention in MPED models is closely connected to the literature of structured output prediction for MLC. Ghamrawi and McCallum (2005) used conditional random fields (Lafferty et al., 2001) to model dependencies among labels and features for MLC by learning a distribution over pairs of labels to input features. In another research direction, recently proposed SPENs (structured prediction energy network (Belanger and McCallum, 2016; Tu and Gimpel, 2018)) and Deep Value Networks (Gygli et al., 2017) tackled MLC by optimizing

different variants of structured loss formulations. In contrast to SPEN and related methods which use an iterative refinement of the output label predictions, our method is a simpler feed forward block to make predictions in one step, yet still models dependencies through attention mechanisms on embeddings. However, we plan to expand MPED models by adding a structured loss formulation.

**Graph Neural Networks (GNNs)** Passing embedding messages from node to neighbor nodes connects to a large body of literature on graph neural networks (Battaglia et al., 2018) and embedding models for structures (Dai et al., 2016). The key idea is that instead of conducting probabilistic operations (e.g., product or re-normalization), the proposed models perform nonlinear function mappings in each step to learn feature representations of structured components. Neural message passing networks (Gilmer et al., 2017), graph attention networks (Veličković et al., 2017) and neural relation models (Battaglia et al., 2016) follow similar ideas to pass the embedding from node to neighbor nodes or neighbor edges. There have been many recent works extending the basic GNN framework to update nodes using various message passing, update, and readout functions (Kipf and Welling, 2016; Hamilton et al., 2017; Duvenaud et al., 2015; Li et al., 2015; Gilmer et al., 2017; Battaglia et al., 2016; Kearnes et al., 2016; Zheng et al., 2018). We refer the readers to (Battaglia et al., 2018) for a survey. However, none of these have used GNNs for MLC.

## 3 Experiments

In the appendix, we have added the details of training/hyperparameters (5.2), datasets (5.1), evaluation metrics (5.3), and explained how we selected baseline models from previous work (5.8). We explain our own MPED variations in 5.7 and previous work baselines in section 5.8.

In short, we compare MPED to (1) MPED Prior $G_{DEC}$ (by using a known label graph), (2) MPED Edgeless $G_{DEC}$ (by removing label-to-label message passing), (3) MPED Autoregressive (by predicting labels sequentially), (4) MP BR (binary relevance MLP output used on the mean of MPNN$_{xx}$ embeddings), and the baselines reported in related works. We compare our models on seven real world datasets, which vary in the number of samples, number of labels, input type (sequential, tabular, graph), and output type (unknown, known label graph), as summarized in Table 3.

Across all datasets, MPED outperforms or achieves similar results as the baseline models. Most importantly, we show that autoregressive models are not crucial in MLC for most metrics, and non-autoregressive models result in a significant speedup at test time.

**Accuracy (across different input and output types)** Table 2 shows the performance of different models across the 7 datasets. Significance across models is shown in Appendix Table 4. For subset accuracy (ACC), autoregressive models perform the best, but at a small margin of increase. However, autoregressive models that predict only positive labels are targeted at maximizing subset accuracy, but they perform poorly on other metrics. For all other metrics, autoregressive models are not essential.

One important observation is that for most datasets, MPED outperforms the autoregressive models in both miF1 (frequent labels) and more importantly, maF1 (rare labels). Since maF1 favors models which can predict the rare labels, this shows that autoregressive models with beam search often make the wrong predictions on the rare labels (which are ordered last in the sequence during training). MPED is a solid choice across all metrics as it comes the closest to subset accuracy as the autoregressive models, but also performs well in other metrics.

While MPED does not explicitly model label dependencies as autoregressive or structured prediction models do, it seems as though the attention weights do learn some dependencies among labels (Visualizations, Tables 3 and 4). This is indicated by the fact that MPED, which uses label-to-label attention, mostly outperforms the ones which don't, indicating that it is learning label dependencies. Table 2 shows 3 time step models results, but a comparison of time steps is shown in Figure 2.

**Speed** Table 2 shows the per epoch train and test times for each model. All models are trained and tested on the same GPU using a batch size of 32. At test time, since the autoregressive model cannot be parallelized, MPED and other non-autoregressive models are significantly faster. During training, the autoregressive model can be parallelized because the true labels are fed as the previous label. Since the autoregressive models only predict the $\rho$ positive labels, they can be faster at training time, whereas the MPED model is predicting the probability for all labels. MPED results in a mean of 1.7x and 5.0x training and testing speedups, respectively, over Seq2Seq autoregressive models.

**Interpretability** We present visualizations of the input-to-label and label-to-label attention weights (averaged across the 4 attention heads) in the Appendix. In the visualizations, we show the positive

| Dataset | Model | ACC | HA | ebF1 | miF1 | maF1 | Average | Train Time | Test Time |
|---|---|---|---|---|---|---|---|---|---|
| Reuters | MPED | 0.834 | 0.997 | **0.902** | 0.871 | 0.503 | **0.822** | 0.788 (1.5x) | 0.116 (2.1x) |
| | MPED Prior $G_{DEC}$ | 0.827 | 0.997 | 0.899 | 0.870 | **0.507** | 0.820 | 0.788 | 0.116 |
| | MPED Edgeless $G_{DEC}$ | 0.828 | 0.997 | 0.898 | **0.873** | 0.495 | 0.818 | 0.738 | 0.111 |
| | MPED Autoregressive | 0.835 | 0.997 | 0.899 | 0.868 | 0.496 | 0.819 | 0.618 | 0.374 |
| | RNN Seq2Seq | **0.837** | 0.996 | 0.900 | 0.861 | 0.496 | 0.818 | 1.187 | 0.242 |
| | RNN Seq2Seq (Nam et al) | 0.828 | 0.996 | 0.894 | 0.858 | 0.457 | 0.807 | - | - |
| | MP BR | 0.824 | 0.997 | 0.891 | 0.870 | 0.463 | 0.809 | 0.595 | 0.102 |
| | BR (Nam et al) | 0.750 | 0.995 | 0.840 | 0.818 | 0.308 | 0.742 | - | - |
| Bibtex | MPED | 0.178 | 0.987 | **0.443** | 0.444 | **0.362** | **0.483** | 0.376 (2.1x) | 0.08 (2.1x) |
| | MPED Prior $G_{DEC}$ | 0.178 | 0.987 | 0.437 | 0.440 | 0.331 | 0.475 | 0.376 | 0.08 |
| | MPED Edgeless $G_{DEC}$ | 0.176 | 0.987 | 0.416 | **0.449** | 0.329 | 0.472 | 0.310 | 0.013 |
| | MPED Autoregressive | **0.197** | 0.984 | 0.432 | 0.419 | 0.315 | 0.470 | 0.206 | 0.208 |
| | RNN Seq2Seq | 0.195 | 0.985 | 0.393 | 0.384 | 0.282 | 0.448 | 0.778 | 0.165 |
| | MP BR | 0.168 | 0.987 | 0.410 | 0.430 | 0.355 | 0.470 | 0.193 | 0.011 |
| | InfNet SPEN (Tu & Gimpel) | - | - | 0.422 | - | - | - | - | - |
| | SPEN (Belanger & McCallum) | - | - | 0.422 | - | - | - | - | - |
| | BR (Belanger & McCallum) | - | - | 0.389 | - | - | - | - | - |
| Delicious | MPED | 0.007 | 0.982 | 0.368 | 0.380 | **0.200** | **0.388** | 3.172 (1.1x) | 0.473 (3.2x) |
| | MPED Prior $G_{DEC}$ | 0.009 | 0.982 | 0.369 | **0.381** | 0.197 | **0.388** | 3.172 | 0.473 |
| | MPED Edgeless $G_{DEC}$ | 0.007 | 0.982 | 0.332 | 0.358 | 0.187 | 0.373 | 2.965 | 0.316 |
| | MPED Autoregressive | **0.014** | 0.973 | 0.322 | 0.325 | 0.167 | 0.360 | 1.158 | 5.129 |
| | RNN Seq2Seq | 0.008 | 0.980 | 0.320 | 0.329 | 0.166 | 0.361 | 3.349 | 1.497 |
| | MP BR | 0.005 | 0.982 | 0.354 | 0.378 | 0.195 | 0.383 | 0.502 | 0.043 |
| | InfNet SPEN (Tu & Gimpel) | - | - | 0.375 | - | - | - | - | - |
| | SPEN (Belanger & McCallum) | - | - | 0.375 | - | - | - | - | - |
| | BR (Belanger & McCallum) | - | - | **0.378** | - | - | - | - | - |
| Bookmarks | MPED | 0.247 | 0.991 | **0.394** | **0.370** | **0.277** | **0.456** | 9.664 (1.2x) | 1.849 (1.3x) |
| | MPED Prior $G_{DEC}$ | 0.246 | 0.991 | 0.392 | **0.370** | **0.277** | 0.455 | 9.664 | 1.849 |
| | MPED Edgeless $G_{DEC}$ | 0.245 | 0.991 | 0.350 | 0.354 | 0.256 | 0.439 | 8.501 | 1.534 |
| | MPED Autoregressive | 0.252 | 0.988 | 0.347 | 0.304 | 0.209 | 0.420 | 7.146 | 5.74 |
| | RNN Seq2Seq | **0.273** | 0.990 | 0.362 | 0.329 | 0.237 | 0.438 | 11.863 | 2.411 |
| | MP BR | 0.241 | 0.991 | 0.382 | 0.364 | 0.275 | 0.451 | 8.022 | 1.312 |
| | InfNet SPEN (Tu & Gimpel) | - | - | 0.376 | - | - | - | - | - |
| | SPEN (Belanger & McCallum) | - | - | 0.344 | - | - | - | - | - |
| | MLP BR (Belanger & McCallum) | - | - | 0.338 | - | - | - | - | - |
| RCV1 | MPED | 0.655 | 0.993 | 0.881 | 0.871 | 0.705 | 0.821 | 98.346 (1.2x) | 1.003 (1.7x) |
| | MPED Prior $G_{DEC}$ | 0.651 | 0.992 | 0.880 | 0.867 | 0.701 | 0.818 | 98.346 | 1.003 |
| | MPED Edgeless $G_{DEC}$ | 0.650 | 0.993 | **0.883** | **0.875** | **0.721** | **0.824** | 90.142 | 0.884 |
| | MPED Autoregressive | **0.660** | 0.992 | 0.881 | 0.868 | 0.715 | 0.823 | 40.366 | 1.928 |
| | RNN Seq2Seq | 0.655 | 0.992 | 0.880 | 0.865 | 0.698 | 0.818 | 119.82 | 1.727 |
| | MP BR | 0.640 | 0.992 | 0.870 | 0.864 | 0.642 | 0.802 | 69.957 | 0.593 |
| | MLP BR (Nam et al) | 0.584 | 0.991 | 0.844 | 0.840 | 0.657 | 0.783 | - | - |
| TFBS | MPED | 0.038 | 0.961 | 0.209 | 0.325 | 0.238 | 0.354 | 187.14 (2.5x) | 13.048 (4.2x) |
| | MPED Prior $G_{DEC}$ | 0.023 | 0.962 | 0.201 | 0.319 | 0.227 | 0.347 | 187.14 | 13.048 |
| | MPED Edgeless $G_{DEC}$ | 0.027 | 0.962 | 0.077 | 0.238 | 0.139 | 0.288 | 126.83 | 12.295 |
| | MPED Autoregressive | 0.057 | 0.963 | 0.201 | **0.354** | **0.245** | 0.364 | 163.04 | 114.87 |
| | MP BR | 0.035 | 0.961 | 0.203 | 0.315 | 0.147 | 0.332 | 111.33 | 7.715 |
| | RNN Seq2Seq | **0.114** | 0.961 | **0.249** | 0.311 | 0.199 | **0.367** | 459.41 | 54.651 |
| SIDER | MPED | 0.007 | 0.748 | **0.766** | **0.795** | **0.667** | **0.597** | 0.027 (2.5x) | 0.003 (21x) |
| | MPED Edgeless $G_{DEC}$ | 0.007 | 0.749 | 0.765 | **0.795** | 0.663 | 0.596 | 0.025 | 0.002 |
| | MPED Autoregressive | 0.007 | 0.755 | 0.719 | 0.771 | 0.602 | 0.571 | 0.025 | 0.137 |
| | MP BR | 0.007 | 0.749 | **0.766** | 0.793 | **0.667** | 0.596 | 0.021 | 0.002 |
| | RNN Seq2Seq | 0.000 | 0.593 | 0.356 | 0.389 | 0.207 | 0.309 | 0.068 | 0.063 |

Table 2: **Results**. Across all 7 datasets, MPED produces similar or better average metric scores to baseline models. MPED results in a mean of 1.7x and 5.0x training and testing speedups, respectively, over the previous state-of-the-art probabilistic MLC method, RNN Seq2Seq. Speedups over RNN Seq2Seq model are shown in minutes per epoch in parentheses for the MPED model. Bold numbers show the best performing method(s).

labels only, and the darker lines show higher attention weights to the corresponding label or word. The attention weights clearly learn certain relationships between input-label pairs as well as the label-label pairs, which is all done in an unsupervised manner. In future work, we plan to add a structured prediction loss function which will likely improve the attention mechanisms and the model's ability to estimate the joint probability.

## 4 CONCLUSION

In this work we present Message Passing Encoder-Decoder (MPED) Networks which achieve a significant speedup at close to the same performance as autoregressive models for MLC. We open a new avenue of using neural message passing to model label dependencies in MLC tasks. In addition, we show that our method is able to handle various input data types (sequence, tabular, graph), as well various output label structures (known vs unknown). One of our future extensions is to adapt the current model to predict more dynamic outputs.

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

# 5 APPENDIX:

| Dataset | Input Type | #Train | #Val | #Test | Labels ($L$) | Mean Labels /Sample | Median Labels /Sample | Max Labels /Sample | Mean Samples /Label | Median Samples /Label | Max Samples /Label |
|---------|-----------|--------|------|-------|--------|------|------|------|-----------|--------|---------|
| Reuters | Sequential | 6,993 | 777 | 3,019 | 90 | 1.23 | 1 | 15 | 106.50 | 18 | 2,877 |
| RCV1 | Sequential | 703,135 | 78,126 | 23,149 | 103 | 3.21 | 3 | 17 | 24,362.15 | 7,250 | 363,991 |
| TFBS | Sequential | 1,671,873 | 301,823 | 323,796 | 179 | 7.62 | 2 | 178 | 84,047.43 | 45,389 | 466,876 |
| BibTex | Bag-of-Words | 4,377 | 487 | 2,515 | 159 | 2.38 | 2 | 28 | 72.79 | 54 | 689 |
| Delicious | Bag-of-Words | 11,597 | 1,289 | 3,185 | 983 | 19.06 | 20 | 25 | 250.15 | 85 | 5,189 |
| Bookmarks | Bag-of-Words | 48,000 | 12,000 | 27,856 | 208 | 2.03 | 1 | 44 | 584.67 | 381 | 4,642 |
| SIDER | Graph | 1,141 | 143 | 143 | 27 | 15.3 | 16 | 26 | 731.07 | 851 | 1185 |

Table 3: Dataset Statistics

| Metric | Significance |
|--------|-------------|
| ACC | RNN Seq2Seq $\succ$ {MPED Edgeless $G_{DEC}$, BR} |
|  | MPED Autoregressive $\succ$ {MPED Edgeless $G_{DEC}$, BR} |
| HA | None |
| ebF1 | MPED $\succ$ {BR, RNN Seq2Seq} |
| miF1 | MPED $\succ$ {RNN Seq2Seq} |
| maF1 | MPED $\succ$ {RNN Seq2Seq} |

Table 4: MLC Model Significance Using the Nemenyi Test (Read et al., 2009; Demšar, 2006)

## 5.1 DATASETS

We test our method against baseline methods on seven different multi-label sequence classification datasets. The datasets are summarized in Table 3. We use Reuters-21578, Bibtex (Tsoumakas et al., 2009), Delicious (Tsoumakas et al.), Bookmarks (Katakis et al.), RCV1-V2 (Lewis et al., 2004), our own DNA protein binding dataset (TFBS) from Consortium et al. (2012), and SIDER (Kuhn et al., 2015), which is side effects of drug molecules. As shown in the table, each dataset has a varying number of samples, number of labels, positive labels per sample, and samples per label. For BibTex and Delicious, we use 10% of the provided training set for validation. For the TFBS dataset, we use 1 layer of convolution at the first layer to extract "words" from the DNA characters (A,C,G,T), as commonly done in deep learning models for DNA.

For datasets which have sequential ordering of the input components (Reuters, RCV1), we add a positional encoding to the word embedding as used in Vaswani et al. (2017) (sine and cosine functions of different frequencies) to encode the location of each word in the sentence. For datasets with no ordering or graph stucture (Bibtex, Delicious, Bookmarks, which use bag-of-word input representations) we do not use positional encodings. For inputs with an explicit graph representation (SIDER), we use the known graph structer.

## 5.2 MODEL HYPERPARAMETERS

We validate our model on seven MLC datasets. These datasets cover a wide spectrum of input data types, including: raw English text (sequential form), bag-of-words (tabular form), and drug molecules (graph form).

For all 6 datasets except SIDER, we use the same MPED model with $T$=3 time steps, $d = 512$, and $K$=4 attention heads. Since SIDER is significantly smaller, we use $T$=1 time step, $d = 64$, and $K$=4 attention heads. We trained our models on an NVIDIA TITAN X Pascal with a batch size of 32. We used Adam (Kingma and Ba, 2014) with betas= $(0.9, 0.999)$, eps=1e-08, and a learning rate of 0.0002 for each dataset. We used dropout of $p = 0.2$ for all models. The MPED models also use layer normalization (Ba et al., 2016) around each of the attention and feedforward layers. The non-autoregressive models are trained with binary cross-entropy on each label and the autoregressive models are trained with cross entropy across all possible labels at each position.

## 5.3 EVALUATION METRICS

Multi-label classification methods can be evaluated with many different metrics which each evaluate different strengths or weaknesses. We use the same 5 evaluation metrics from Nam et al. (2017).

All of our autoregressive models predict only the positive labels before outputting a stop signal. This is a special case of PCC models (explained as PCC+ in section 5.4), which have been shown to outperform the binary prediction of each label in terms of performance and speed. These models use beam search at inference time with a beam size of 5. For the non-autoregressive models, to convert the labels to $\{0, 1\}$ we chose the best threshold on the validation set from the same set of thresholds used in Tu and Gimpel (2018).

**Example-based measures** are defined by comparing the target vector $\boldsymbol{y}$ to the prediction vector $\hat{\boldsymbol{y}}$. Subset Accuracy (ACC) requires an exact match of the predicted labels and the true labels: $\text{ACC}(\boldsymbol{y}, \hat{\boldsymbol{y}}) = \mathbb{I}[\boldsymbol{y} = \hat{\boldsymbol{y}}]$. Hamming Accuracy (HA) evaluates how many labels are correctly predicted in $\hat{y}$: $\text{HA}(\boldsymbol{y}, \hat{\boldsymbol{y}}) = \frac{1}{L} \sum_{j=1}^{L} \mathbb{I}[y_j = \hat{y}_j]$. Example-based F1 (ebF1) measures the ratio of correctly predicted labels to the sum of the total true and predicted labels: $\frac{2 \sum_{j=1}^{L} y_j \hat{y}_j}{\sum_{j=1}^{L} y_j + \sum_{j=1}^{L} \hat{y}_j}$.

**Label-based measures** treat each label $y_j$ as a separate two-class prediction problem, and compute the number of true positives ($tp_j$), false positives ($fp_j$), and false negatives ($fn_j$) for a label. Macro-averaged F1 (maF1) measures the label-based F1 averaged over all labels: $\frac{1}{L} \sum_{j=1}^{L} \frac{2tp_j}{2tp_j + fp_j + fn_j}$. Micro-averaged F1 (miF1) measures the label-based F1 averaged over each sample: $\frac{\sum_{j=1}^{L} 2tp_j}{\sum_{j=1}^{L} 2tp_j + fp_j + fn_j}$. High maF1 scores usually indicate high performance on less frequent labels. High miF1 scores usually indicate high performance on more frequent labels.

## 5.4 BACKGROUND OF MULTI-LABEL CLASSIFICATION:

MLC has a rich history in text (McCallum; Ueda and Saito, 2003), images (Tsoumakas and Katakis, 2006; Elisseeff and Weston, 2002), bioinformatics (Tsoumakas and Katakis, 2006; Elisseeff and Weston, 2002), and many other domains. MLC methods can roughly be broken into several groups, which are explained as follows.

Label powerset models (LP) (Tsoumakas and Vlahavas, 2007; Read et al., 2011), classify each input into one label combination from the set of all possible combinations $\mathcal{Y} = \{\{1\}, \{2\}, ..., \{1, 2, ..., L\}\}$. LP explicitly models the joint distribution by predicting the one subset of all positive labels. Since the label set $Y$ grows exponentially in the number of total labels ($2^L$), classifying each possible label set is intractable for a modest $L$. In addition, even in small $L$ tasks, LP suffers from the "subset scarcity problem" where only a small amount of the label subsets are seen during training, leading to bad generalization.

Binary relevance (BR) methods predict each label separately as a logistic regression classfier for each label (Zhang and Zhou, 2005; Godbole and Sarawagi, 2004). The naïve approach to BR prediction is to predict all labels independently of one another, assuming no dependencies among labels. That is, BR uses the following conditional probability parameterized by learned weights $W$:

$$P_{BR}(Y|X; W) = \prod_{i=1}^{L} p(Y_i | X_{1:S}; W) \tag{21}$$

Probabilistic classifier chain (PCC) methods (Dembczynski et al., 2010; Read et al., 2009) are autoregressive models that estimate the true joint probability of output labels given the input by using the chain rule, predicting one label at a time:

$$P_{PCC}(Y|X; W) = \prod_{i=1}^{L} p(Y_i | Y_{1:i-1}, X_{1:S}; W) \tag{22}$$

Two issues with PCC models are that inference is very slow if $L$ is large, and the errors propagate as $L$ increases (Montañes et al., 2014). To mitigate the problems with both LP and PCC methods, one solution is to only predict the true labels in the LP subset. In other words, only predicting the positive labels (total of $\rho$ for a particular sample) and ignoring all other labels, which we call PCC+. Similar to PCC, the joint probability of PCC+ can be computed as product of conditional probabilities, but unlike PCC, only $\rho < L$ terms are predicted as positive:

$$P_{PCC+}(Y|X; W) = \prod_{i=1}^{\rho} p(Y_{p_i} | Y_{p_{1:i-1}}, x_{1:S}; W) \tag{23}$$

This can be beneficial when the number of possible labels $L$ is large, reducing the total number of prediction steps. However, in both PCC and PCC+, inference is done using beam search, which is a costly dynamic programming step to find the optimal prediction.

MPED methods approximate the following factored formulation, where $\mathcal{N}(Y_i)$ denotes the neighboring nodes of $Y_i$.

$$P_{G2G}(Y|X; W) = \prod_{i=1}^{L} p(Y_i | \{Y_{\mathcal{N}(Y_i)}\}, X_{1:S}; W). \tag{24}$$

## 5.5 SEQ2SEQ MODELS

In machine translation (MT), sequence-to-sequence (Seq2Seq) models have proven to be the superior method, where an encoder RNN reads the source language sentence into an encoder hidden state, and a decoder RNN

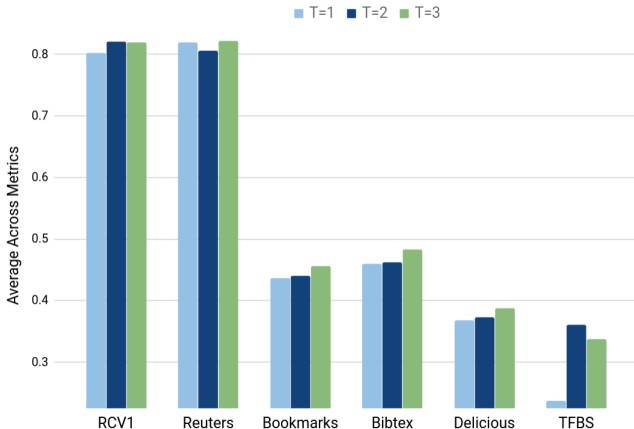

Figure 2: Average Across Metrics for $T$=1, $T$=2, and $T$=3 $G_{DEC}$ time steps. In these experiments, the encoder $G_{ENC}$ is processed with a fixed $T$=3 time steps, and the decoder time steps are varied. We do not compare time steps for the SIDER dataset since it is too small, and we only evaluate using $T$=1.

translates the hidden state into a target sentence, predicting each word autoregressively (Sutskever et al., 2014). Bahdanau et al. (2014) improved this model by introducing "neural attention" which allows the decoder RNN to "attend" to every encoder word at each step of the autoregressive translation.

Recently, Nam et al. (2017) showed that, across several metrics, state-of-the-art MLC results could be achieved by using a recurrent neural network (RNN) based encoder-to-decoder framework for Equation 23 (PCC+). They use a Seq2Seq RNN model (Seq2Seq Autoregressive) which uses one RNN to encode $x$, and a second RNN to predict each positive label sequentially, until it predicts a 'stop' signal. This type of model seeks to maximize the 'subset accuracy', or correctly predict every label as its exact 0/1 value.

Vaswani et al. (2017) eliminated the need for the recurrent network in MT by introducing the Transformer. Instead of using an RNN to model dependencies, the Transformer explicitly models pairwise dependencies among all of the features by using attention (Bahdanau et al., 2014; Xu et al.) between signals. This speeds up training time because RNNs can't be fully parallelized but, the transformer still uses an autoregressive decoder.

## 5.6 Drawbacks of Autoregressive Models

Autoregressive models have been proven effective for machine translation and MLC (Sutskever et al., 2014; Bahdanau et al., 2014; Nam et al., 2017). However, predictions must be made sequentially, eliminating parallelization. Also, beam search is typically used at test time to find optimal predictions. But beam search is limited by the time cost of large beams sizes, making it difficult to optimally predict many output labels (Koehn and Knowles, 2017).

In addition to speed constraints, beam search for autoregressive inference introduces a second drawback: initial wrong predictions will propagate when using a modest beam size (e.g. most models use a beam size of 5). This can lead to significant decreases in performance when the number of positive labels is large. For example, the Delicious dataset has a median of 19 positive labels per sample, and it can be very difficult to correctly predict the labels at the end of the prediction chain.

Autoregressive models are well suited for machine translation because these models mimic the sequential decoding process of real translation. However, for MLC, the output labels have no intrinsic ordering. While the joint probability of the output labels is independent of the label ordering via autoregressive based inference, the chosen ordering can make a difference in practice (Vinyals et al., 2015; Nam et al., 2017). Some ordering of labels must be used during training, and this chosen ordering can lead to unstable predictions at test time.

Our non-autoregressive version connects to Gu et al. (2017) who removed the autoregressive decoder in MT with the Non-Autoregressive Transformer. In this model, the encoder makes a proxy prediction, called "fertilities", which are used by the decoder to predict all translated words at once. The difference between their model and ours is that we have a constant label at each position, so we don't need to marginalize over all possible labels at each position.

## 5.7 Model Variations

**MPED**: In the full MPED model, we use 3 encoder time steps and 3 decoder time steps with node to node attention in both the encoder and decoder graphs, and $K$=4 attention heads.

**MPED Prior $G_{DEC}$**: The same as MPED except that we use a prior representation of $G_{DEC}$ rather than fully connected. For Reuters, Bibtex, Delicious, and Bookmarks we have extracted label graphs using label similarity (using fixed WUP scores) from WordNet. For RCV1, we use the topological graph from the RCV1 dataset. For TFBS, we use String-DB (Szklarczyk et al., 2016) to obtain the label graph representing TF-TF protein interactions.

**MPED Edgeless $G_{DEC}$**: In this baseline, we remove MPNN$_{yy}$, and only use MPNN$_{xy}$ to pass messages from the inputs to the labels, and not between labels. This model assumes no dependencies among output labels.

**MPED Autoregressive:** Instead of predicting all labels simultaneously and modelling label dependencies using graph attention, the MPED Autoregressive variation models $p(y_{i+1}|y_{1:i}, x_{1:S}; W)$ by predicting each positive label $y_i$ sequentially and using attention on $\{c_{1:S}^T\}$ and the previously predicted labels $(\hat{y}_1, ..., \hat{y}_i)$ until a stop signal is predicted (see PCC+ in Appendix). While it is possible to predict the probability for each label (i.e. regular PCC), this is extremely time consuming when the number of possible labels is large (and in general, doesn't increase performance (Nam et al., 2017)).

**BR**: The mean node output of a MPNN$_{xx}$ is used as the input to a 2 layer multilayer perceptron (MLP) readout module. This model assumes no dependencies among output labels.

**RNN Seq2Seq**: A 3-layer RNN Seq2Seq model with attention. This is similar to RNN Seq2Seq from Nam et al. (2017) (RNN$^m$ in their paper), except with 3 layers.

## 5.8 BASELINE COMPARISONS

For Reuters and RCV1-V2, we compare against the MLP and autoregressive model used in (Nam et al., 2017), which use these two datasets. **MLP**: standard MLP without the self attention input representation. **RNN Seq2Seq**: RNN PCC+ autoregressive model from (Nam et al., 2017), which is a GRU encoder/decoder model with pretrained word embeddings. We omit the binary relevance RNN since it was shown that it doesn't perform well and is extremely slow.

For BibTex, Delicious, Bookmarks, we compare against the models from Belanger and McCallum (2016) and Tu and Gimpel (2018), which use these three datasets. They only evaluate using ebF1, so this is the only metric for which we compare our model with theirs. **MLP** (Belanger and McCallum, 2016): Standard 2-layer MLP. **SPEN** (Belanger and McCallum, 2016): original structured prediction energy network. **InfNet SPEN** (Tu and Gimpel, 2018): improved SPEN model, which uses an inference network to estimate $\hat{y}$.

## 5.9 VISUALIZATIONS

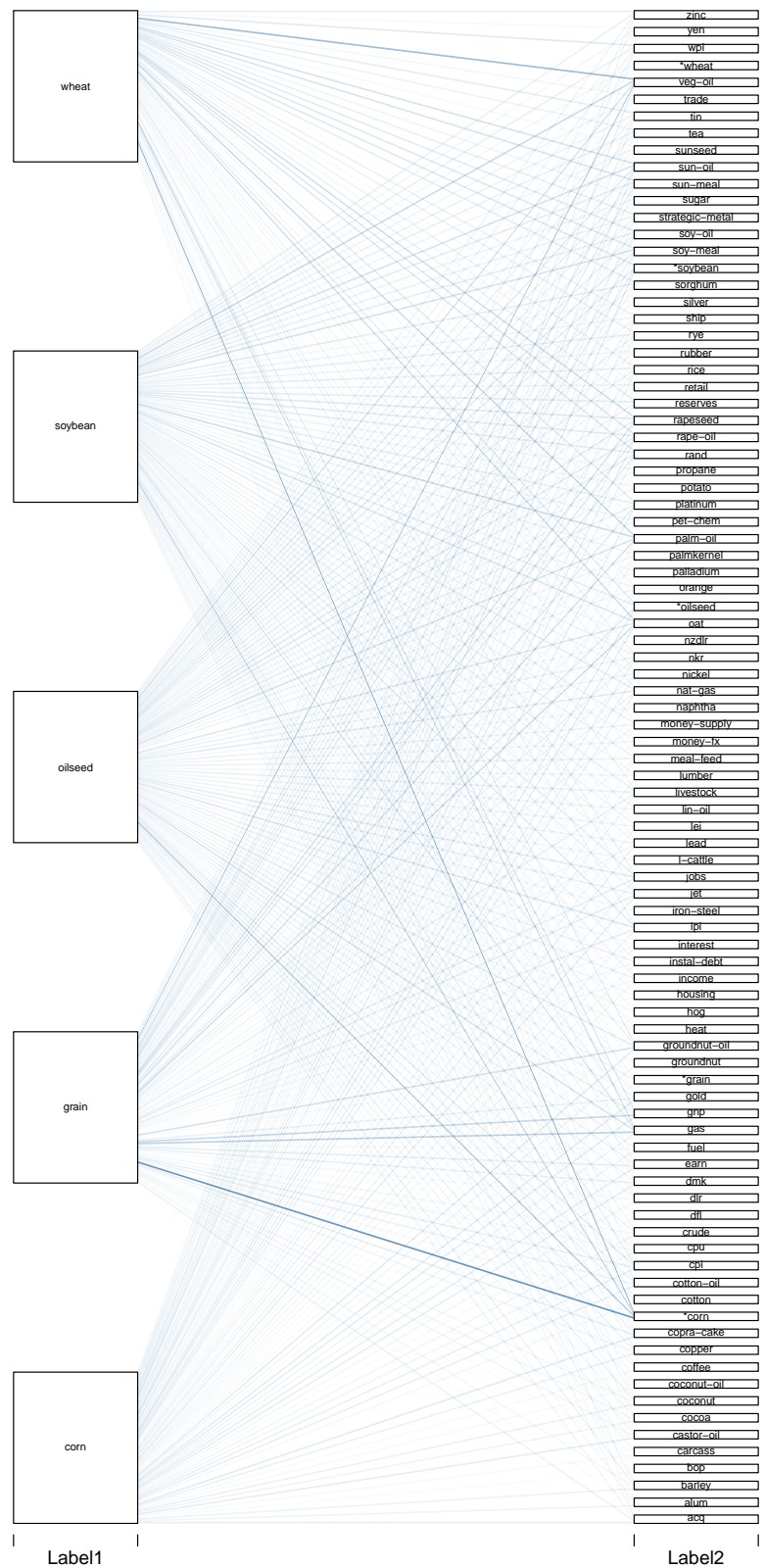

Figure 3: Label-To-Label Attention Weights. On the left are the positive labels, and on the right are all labels. These are taken from the layer 2 Label-To-Label attentions on a sample from Reuters-21578 (same input sample as shown in 4.

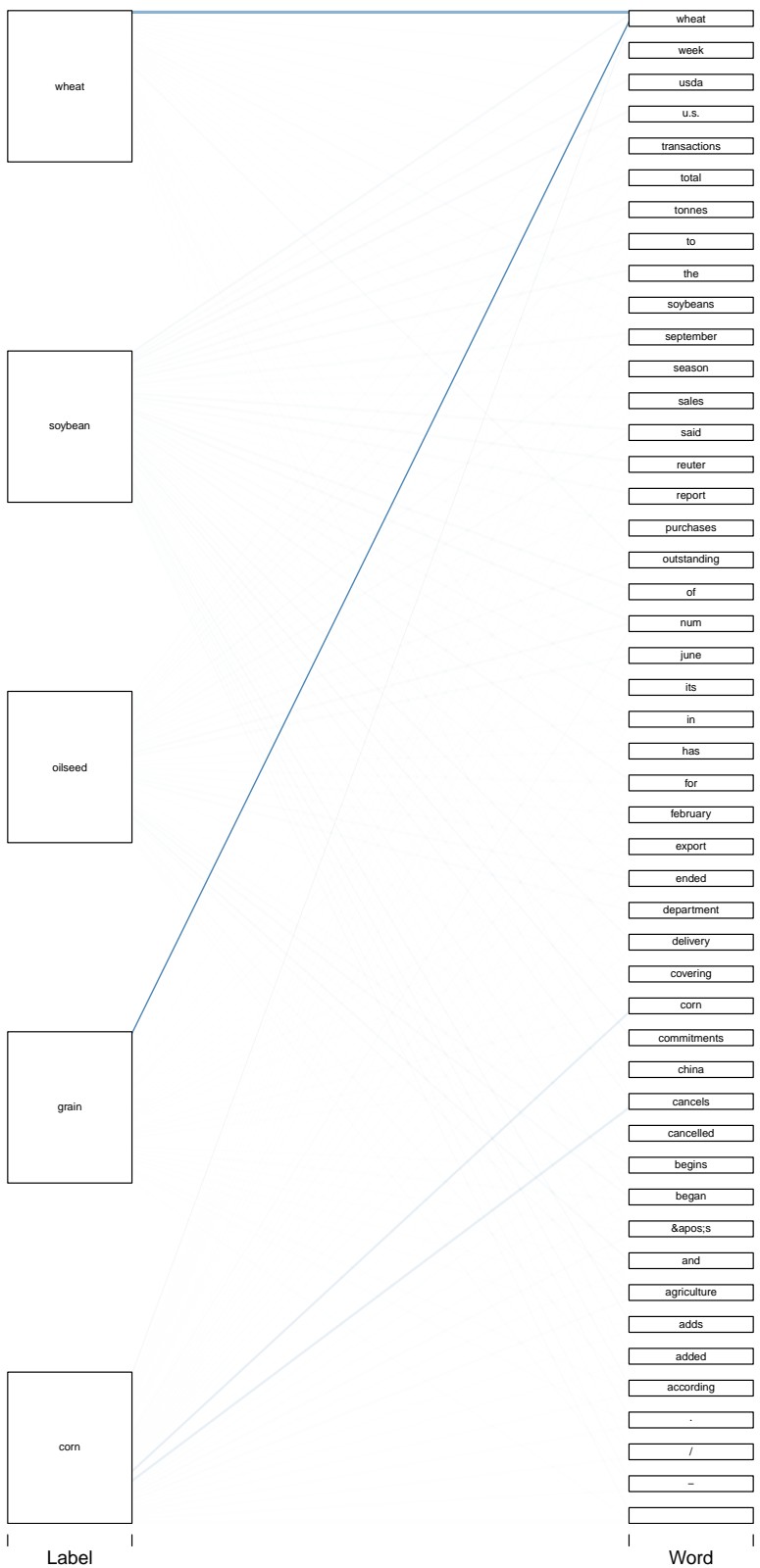

Figure 4: Input to Label Attention Weights. On the left are the positive labels, and on the right are all input components (words). These are taken from the layer 1 Input to Label attentions on a sample from Reuters-21578.

