# OpenReview forum: "Neural Message Passing for Multi-Label Classification"
_ICLR.cc/2019/Conference_

### Official Review · AnonReviewer2 · 2018-10-29
**Graph2Graph without any graph structured inputs or outputs.**

**Rating:** 5
**Confidence:** 2

**Review:**

This paper proposes an encoder-decoder model based on the graph representation of inputs and outputs to solve the multi-label classification problem. The proposed model considers the output labels as a fully connected graph where the pair-wise interaction between labels can be modelled.

Overall, although the proposed approach seems interesting, the representation of the paper needs to be improved. Below I listed some comments and suggestions about the paper.

- The proposed model did not actually use any graph structure of input and output, which can potentially mislead the readers of the paper. For instance, the encoder is just a fully connected feed-forward network with an additional attention mechanism. In the same sense, the decoder is also just a fully connected feed-forward network. Furthermore, the inputs and outputs used throughout the paper do not have any graph structure or did not use any inferred graph structure from data. I recommend using any graph-structured data to show that the proposed model can actually work with the graph-structured data (with proper graph notations) or revise the manuscript without graph2graph representation.

- I personally do not agree with the statement that the proposed model is interpretable because it can visualise the relation between labels through the attention. NN is hard to interpret because the weight structure cannot be intuitively interpretable. In the same sense, the proposed model cannot avoid the problem with the nature of black-box mechanism. Especially, multiple weight matrices are shared across the different layers, which makes it more difficult to interpret. Although the attention weights can be visualised, how can we visualise the decision process of the model from end-to-end? The question should be answered to claim that the model is interpretable.

- 2.2.1, 2.2.2, 2.3 shares the similar network layer construction, which can be represented as a new layer of NN with different inputs (or at least 2.2.2 and 2.3 have the same layer structure). It would be better to encapsulate these explanations into a new NN module which can be reused multiple parts of the manuscript for a concise explanation.

- Although the network claims to model the interactions between labels, the final prediction of labels are conditionally independent to each other, whereas the energy based models such as SPEN models the structure of output directly. In that sense, the model does not take into account the structure of output when the prediction is made although the underlying structure seems to model the 'pair-wise' interaction between labels.

- In Table1, if the bold-face is used to emphasise the best outcome, I found it is inconsistent with the result (see the output of delicious and tfbs datasets).

- Is it more natural to explain the encoder first followed by the decoder?

---

> ### Author Response · Authors · 2018-11-21
> **Are there errors in the bold font of the table indicating best performing methods?**
>
> We would like to thank reviewer 2 for pointing out this critical typo. This has since been updated.

---

> ### Author Response · Authors · 2018-11-21
> **Can the model explanation be encapsulated and made more clear? And should the encoder be explained first?**
>
> We thank reviewer 2 for recommending this way to make our paper more concise easier to understand. We agree that this is a much better way to explain our method. We have since updated the manuscript to reflect such an encapsulated approach, as well as incorporated the recommendation to introduce the encoder first.

---

> ### Author Response · Authors · 2018-11-21
> **Is the model interpretable?**
>
> We would like to thank reviewer 2 for asking an important question about the importance of the attention mechanisms. There are a few reasons behind our claim:
> 1. We make this claim as an extension of previous works using attention (Bahdanau et al. 2015, Vaswani et al. 2017, Velickovic et al. 2017) which all show the advantage of interpretability of the attention weights on natural language and graph networks.
> 2. There exists a large literature (Simonyan et al., 2014, Bach et al. 2015, Tulio Ribeiro et al. 2016) about visualizing deep neural networks, and there are different ways to categorize these types of visualizations. Roughly, these lines of work fall into one of the following categories. First, feature attribution methods are concerned with how features of a sample contribute to a model output. Second, interaction attribution methods are concerned with how non-additive effects between features influence an outcome variable. Third, the locally interpretable model-agnostic explanations (LIME - Tulio Ribeiro et al. 2016) approximate model predictions in the local vicinity of a data sample.
>
> Our model provides three different levels of attention weights. The first is our input-to-input attention weights which are concerned with how these non-additive interactions among components contribute the representation of the inputs. In the second level, the input-to-label attention weights are about how the input components differently contribute to the representation of the output labels. In the third level, the label-to-label attention weights are concerned with how non-additive interactions among labels contribute the final representation of the labels before label classification. We agree that our attention weights are not about how features or feature interactions contribute directly to the outcome. We plan to combine our attention weights with approaches such as LIME to provide such end-to-end explanations.

---

> ### Author Response · Authors · 2018-11-21
> **Can this work on graph-structured data?**
>
> We agree that our original explanation was unclear and hard to read. The original model name Graph2Graph was a stretch since we didn’t use explicit graphs in the inputs or outputs. We have done a major revision of our draft summarized as follows:
> 1. We have changed our method name from “Graph2Graph Networks” to “Message Passing Encoder-Decoder (MPED) Networks”. We want to emphasize that our model has not changed at all. We simply revised the presentation of our approach from the graph angle to neural message passing. We feel the current writing is more intuitive, concise, and easier to follow.
> 2. We have added new experiments using graph-structured inputs with a drug side effect dataset which uses drug molecule (graph) inputs.
> 3. In addition, we have added known structure of graph labels, which shows our method can in fact work on known graphs, as explained in the method and experiments section.

---

> ### Author Response · Authors · 2018-11-21
> **Is this a fully connected feed-forward network?**
>
> We would like to thank reviewer 2 for bringing to light an important question about clarity in our explanation, which he have thus revised in the manuscript. We have revised the model figure (Figure 1) into a more intuitive way to show message passing between inputs, from inputs to labels, and between labels.
>
> In Equations 1 and 2 we introduce a generic form of neural message passing, and we show how we implement neural message passing using graph attention in Equations 3-9. This is largely different from how we introduced graph attention in the previous version of our draft, which we think is now more intuitive and straightforward to understand. In summary,
> + Each node is represented as a weighted summation of the messages passed from all of its neighbors.
> +The edges in our Figure 1 are not representing fully connected MLP edges.
> + We have revised Figure 1 to connect better to the equations. For example, in the decoder, the edge weights e_12 from node 2 to node 1 are calculated using Equations 3 and 4. Equation 5 further normalizes e_12, e_13, e_14, e_15 into a_12, a_13, a_14, a_15. a_12 is then used in Equation 6 to weight the messages from node 2 to node 1. Equation 7 aggregates all neighbors’ messages to node 1 by summing over messages from node 2 to 1, node 3 to 1, node 4 to 1, and node 5 to 1. Finally, the aggregated messages from the neighbors of node 1 are used to update the state of node 1 in Equations 8 and 9.
> + The above process (Equations 3-9) is certainly not a fully connected feed-forward network. It can be viewed as 1-dimensional convolution with kernel and stride sizes of 1. This is a key aspect of message passing neural networks, where feature dependencies are learned in a order-invariant manner. It is important to note that the W matrices are shared across node embeddings. In other words the W matrices are not fully connected across all labels and inputs.
>
> We explain how we use attention message passing for MLC using our encoder-decoder approach in Equations 10-16.

---

> ### Author Response · Authors · 2018-11-21
> **How does our model compare to SPEN?**
>
> We thank reviewer 2 for bringing up the important differences between our method and SPEN models. Indeed, we do not explicitly model the output structure as done in SPEN models. In contrast to SPEN and related CRF methods which use an iterative refinement of the output label predictions, our method is a simpler feedforward block to make predictions in one step. However, we plan to expand on our method by adding a SPEN output in future work.

---

### Official Review · AnonReviewer1 · 2018-11-02
**A paper, with a misleading title, presenting experimental results for which statistical significance is not reported.**

**Rating:** 6
**Confidence:** 4

**Review:**

As a reviewer I am expert in learning in structured data domains. Because of that I completely disagree that the proposed title of the paper is not misleading. In fact, both the input and the output of the proposed system are not graphs. Moreover, the intermediate representations are always complete graphs, so there is no graph to graph transformation here. It is the internal topology of the encoder and decoder that corresponds to a complete graph and not the nature of the processed data.
The main intended contribution of the paper is to define a system able to capture the dependencies among input features as well as output labels, so to improve the multi-label classification task addressed by the system. This is obtained by defining a recurrent model with a complete graph topology to both encode the input and decode the output. The decoding part starts from the assumption of independence among the output labels and then, via interaction with the encoded representation of the input, eventually turns to an output where relevant statistical dependences among output labels emerge with decoding. Since both encoding and decoding are recurrent models (with no enforced guarantee to have stable points), the paper proposes to unfold the recursion for a fixed predefined number of time steps.
Presentation of the proposal is generally good, although there are some issues that are not clear. For example, the same weights indices are used for matrices belonging to the encoding and decoding, making the reader to believe that such matrices are shared. In addition, the sentence about model parameters at page 5 is a bit ambiguous and it is not sufficient to resolve the presentation problem.
The discussion at the end of page 4 on the fact that a sequential representation for the input components is not natural is actually out of place for the specific application task selected for presentation. In fact, words in a sentence have an order. The fact that such order is lost with the bag-of-word representation is a problem of preprocessing, not of the nature of the data. In general, however, it is true that forcing an order is not natural.
Going in the merit of the proposal, the number of parameters for the decoder scales quadratically with the number of output labels (fully connected graph). In domains with a large numbers of labels (e.g. thousands) there may be concerns on two different aspects: i) computational burden may grow significantly even if the average number of labels per item is small; ii) proper propagation of information on dependencies among labels may require to use a large value for T (graph hops), i.e. there is a dependency between size of label graph and "useful" value for T.  On this issue, by the way, figures 3 and 4 seem to report incongruent results since, because of symmetries in the model topology, equal and  reciprocal influences between input components (and output labels) would have been expected, but these are not observed in the figures.
Analogous considerations could be done for the encoder when the size of the input is large.
Concerning experimental results, no statistical significance test is performed, so it is not clear to me if the shown improvements are actually significant. Speed-up in training and testing seem at least to give some advantage with respect to other competing approaches, however the scaling problem described above for the decoder (and encoder) may lead to much worst performances in those special cases.
The addressed problem is covered by a large literature, involving many different approaches. It would have been nice to report, for the selected datasets, the best performance (and computation times) obtained by, for example,  probabilistic graphical models or SVM-based models.
The paper seems to refer most of the relevant recent neural-based approaches.
I think the paper is relevant for ICLR (although there is no explicit analysis of the obtained hidden representations) and of interest for a good portion of attendees.

Minor issues:
- two rows before Section 2.2.1: \mathbb{h}_*^2  should be \mathbb{h}_*^1
- equations 4, 5, 9, 10, 14: matrices W are indexed in such a way to assume that each input word/label is associated to a different matrix (i.e., set of parameters). Is this really the case ? How is then managed the fact that different inputs may have a different number of components ? how is a specific matrix assigned to a specific word ? I guess this is a presentation mistake, otherwise there are relevant issues that are completely not addressed by the presentation.
- equation (10): since the output should be interpreted as a probability, why not using a softmax? sigmoidal units by themselves do not guarantee that the outputs sum to 1. I guess you do not have this problem because you adopt batch normalisation. This however is conceptually not nice since there is no uniformity across the dataset. Moreover, the softmax function has a nice probabilistic interpretation in the family of the exponential distributions.
- "[...] we use add a positional encoding..."
- Multi-head Attention: apart for the not so clear description, the equation involving the softmax is missing.
- "[..] the the attention and feedforward layers."
- "[..] the the sum of the total true..."

---

> ### Author Response · Authors · 2018-11-21
> **Minor issues**
>
> We thank reviewer 1 for providing a detailed analysis of minor, yet important issues. There are not separate matrices W for each word/label. They are shared across words or labels, and we have updated the manuscript to reflect this. We have also fixed the mentioned typos and errors.

---

> ### Author Response · Authors · 2018-11-21
> **Significance tests**
>
> We thank reviewer 1 for mentioning statistical significance tests, which were missing from our experiments previously. We used the Nemenyi test, as used in Read et al., 2009. We have added the results to the Appendix Table 4.

---

> ### Author Response · Authors · 2018-11-21
> **Computational burdens of fully connected graphs are a drawback**
>
> We would like to thank reviewer 1 for the detailed analysis of our approach in regard to computational burdens. We agree that this is a drawback that the parameters of our fully connected graphs scale quadratically with the number of output labels. This is one of the future work directions we are working on now. A few possible solutions include:
> 1. Using some sort of hierarchical representation over the label graph.
> 2. Message passing could be restricted to considering only k neighbors instead of all of the neighbors. In this case, some sorting algorithm would need to be used to rank neighbors.
>
> We are unclear about the question regarding Figures 3 and 4. Can you please expand on the incongruent results between the input-to-label weights vs the label-to-label weights?

---

> ### Author Response · Authors · 2018-11-21
> **Representing sentences using fully connected graphs is not natural**
>
> We agree that our wording on representing sentences as a fully connected graph was misleading, which we have since changed. Using a fully connected graph to model the interactions between input components allows us to (1) capture non-local feature interactions, and (2) speed up training and testing time by parallelization.
>
> In addition, our framework is able to model more complex input representations such as chemical molecule samples. We have added one more real world MLC dataset which uses drug molecule inputs (results added in Table 2). In this case, the encoder now works on a known graph instead of a fully connected graph. On the equation level, we only need to change the summation indices (in Equation 10) to reflect the known neighbors of a node.

---

> ### Author Response · Authors · 2018-11-21
> **Unclear notations of weight matrices W**
>
> We thank reviewer 1 for pointing out unclear notations. We have updated the paper to reflect the important point that weights are not shared between the different encoding and decoding modules of our framework. Specifically: weights for input-to-input are represented by W_xx (Equations 10 and 11), weights for input-to-label are represented by W_xy (Equations 12 and 13), weights for input-to-label are represented by W_yy (Equations 14 and 15).

---

> ### Author Response · Authors · 2018-11-21
> **Title and model name were misleading**
>
> We would like to thank reviewer 1 for clarifying this important aspect of our paper. Accordingly, we have thus revised the manuscript with the following changes:
> 1. We have changed our title from “Graph2Graph Networks for Multi-Label Classification” to “Neural Message Passing for Multi-Label Classification”
> 2. We have changed our method name from Graph2Graph Networks to Message Passing Encoder-Decoder (MPED) Networks for MLC
> 3. We have revised the model figure (Figure 1.) into a more intuitive way to show message passing between inputs, from inputs to labels, and between labels.
> 4. We have added new experiments using explicit graph representations of labels, as explained in the method section. We found that modelling the labels using fully connected graphs produces better results in most cases.

---

### Official Review · AnonReviewer3 · 2018-11-06
**Interesting but weaker novelty/experiments/writing**

**Rating:** 4
**Confidence:** 4

**Review:**

The paper describes an approach for using graph neural networks (GNN) to perform multi-label classification (MLC). The main idea is to use attentional pooling to project an input graph into a "label graph", whose nodes correspond to labels on some MLC problem. Multiple rounds of self-attention/message-passing hops can be performed on the input graph and label graph. Each output label is binary-valued, and is predicted from its corresponding node in the label graph. They evaluate on 6 multi-label sequence classification datasets, and report strong perform over baselines.

Though interesting, I recommend rejection for several reasons:

1) The technical contribution has limited novelty. One (very recent) reference this paper misses is "Hierarchical Graph Representation Learning with Differentiable Pooling" by Ying et al. (2018), which uses a very similar mechanism. The field is moving quickly, so references get missed sometimes, however from what I can tell, the graph-coarsening idea presented here isn't that technically distinct from Ying el al.'s. The Mrowca et al. (2018) "Flexible Neural Representation for Physics Prediction" is also fairly similar and should probably at least be cited.

2) There aren't strong baselines. This approach is based on GNNs, and the Graph2MLP results, which is similar to previous GNN graph-level classification methods, are fairly strong too. My suspicion is that with some more tuning and tweaking, the results here would be similar to those of Ying et al., Velickovic et al. (2017)'s Graph Attention Nets, and other models which use what Gilmer et al. (2017) terms the "readout" function for MLC. Without testing some of these other approaches, how can readers be sure this is approach has value over other approaches? The reviews by Gilmer et al. (2017) and Battaglia et al. (2018) summarize a bunch of alternatives that could be tried, some of which use similar encoder/decoder setups (not with the attentional pooling, however, as far as I know).

3) The writing is fairly dense for what is a fairly straightforward idea. And the paper is over 8.5 pages, with key details in the Appendix.

I believe this approach could be quite powerful, and there was clearly a lot of excellent work that went into this project. But because the GNN area is very active, the bar is high. With a little more innovation on the model side (can the same core model be useful for things beyond MLC as well? I'm guessing it could), better baselines, better scholarship, and condensing the writing, I think this paper can be an important step forward.

---

> ### Author Response · Authors · 2018-11-21
> **Can the same core model be useful for things beyond MLC as well?**
>
> We thank reviewer 3 for asking this important question. In general, our message passing encoder-decoder method is a generic framework for any encoder-decoder model where the input and outputs can be represented as graphs. As explained in the updated version of our paper, this could be applied to other input or outputs such as drug molecule inputs (which we have since added). However, we want to show that message passing is a good method for MLC. There are many applications of similar models, for example a parrallel submission in ICLR 2019, the Graph Transformer (https://openreview.net/forum?id=HJei-2RcK7&noteId=HJei-2RcK7).

---

> ### Author Response · Authors · 2018-11-21
> **Writing is fairly dense and could be explained in under 8.5 pages**
>
> We thank reviewer 3 for pointing out this drawback of our explanation. We have since updated the manuscript to reflect such our approach using an encapsulated representation of message passing modules and it now fits in under 8 pages.

---

> ### Author Response · Authors · 2018-11-21
> **Baselines**
>
> We thank reviewer 3 for bringing up an important point about baselines.
> 1. The novelty of our approach is about applying message passing neural networks for MLC. In our experiments, we use state-of-the art MLC baselines including Seq2Seq MLC, SPEN MLC, and Binary Relevance MLC.
> 2. Our message passing method is fundamentally similar to Velickovic et al. 2017 who used attention message passing for graph classification. Our contribution is not about a novel graph neural network, but rather it is a novel approach for MLC. Our current choice of graph attention based neural message passing is able to specify different weights to different nodes in a neighborhood, without requiring any kind of costly matrix operation such as inversion (Kipf & Welling, 2016) or depending on knowing the graph structure a priori. Adding variations of graph neural networks will certainly enrich our paper, which is one of our future, for instance a hierarchical graph representation for our decoder.

---

> ### Author Response · Authors · 2018-11-21
> **Novelty of our paper**
>
> We thank reviewer 3 for the comments.  We have updated our manuscript to provide a clearer explanation of our motivation and method. However, we have the following confusions.
> 1. We would like to ask for a clarification on the graph-coarsening idea, since our method does not use graph coarsening.
> 2. We don’t believe our paper is related to Ying et. al who develop a pooling method for graph classification. Our paper is not for graph classification at all.
> 3. Mrowca et al. introduce a hierarchical representation of node states for state prediction. We believe this is a perpendicular research direction and the  Mrowca et al. paper is not about multi-label classification. Although their task is different than ours, a similar hierarchical representation could be added our model in future work. We have added this our related work section.
> 4. Most importantly, we do not claim novelty in our graphical neural network methodology. Rather, our method is a novel approach for MLC using neural message passing through an encoder-decoder architecture.

---

### Author Response · Authors · 2018-11-21
**Summary of Changes**

We would like to thank our reviewers for providing valuable comments and questions. Please see our revised manuscript, which we have updated to reflect the following changes.

+ We have changed our title from “Graph2Graph Networks for Multi-Label Classification” to “Neural Message Passing for Multi-Label Classification”

+ We have changed our method name from Graph2Graph Networks to Message Passing Encoder-Decoder (MPED) Networks for MLC

+ We have revised the presentation of our approach from the graph angle to neural message passing. We feel the current writing is more intuitive, concise, and easier to follow. We have revised the model figure (Figure 1) into a more intuitive way to show the modular components for message passing between inputs, from inputs to labels, and between labels. In addition, we have revised the writing to reflect this modular component approach of our method. We want to emphasize that our model has not changed at all.

+ We have addressed the helpful reviews about not using known input graph structure by adding a new dataset, SIDER, which is an MLC dataset for predicting multiple side effects using the molecule structure (graph) of a drug. Now our experiments contain seven real world MLC datasets which cover a wide spectrum of input data types, including: raw English text (sequential form), bag-of-words (tabular form), and drug molecules (graph form).

+ We also addressed the concerns about not using known output graphs by expanding our experiments and considering the known graph structure among labels for six datasets. For Reuters, Bibtex, Delicious, and Bookmarks we have extracted label graphs using label similarity from WordNet. For RCV1, we use the known topological graph from the RCV1 dataset. For TFBS, we use String-DB to obtain the label graph representing TF-TF protein interactions. This new set of results are added using MPED Prior G_DEC in Table 2.


We have addressed each reviewer's individual points in separate comments.

---

### Meta-Review · Area_Chair1 · 2018-12-14
**Not a clear acceptance**

**Confidence:** 3
**Recommendation:** Reject

**Metareview:**

The reviewers highlighted aspects of the work that were interesting, particularly on the chosen topic of multi-label output of graph neural networks. However, no reviewer was willing to champion the paper, and in aggregate all reviewers trend towards rejection.